# The Influence of Sonographer Experience on Skeletal Muscle Image Acquisition and Analysis

**DOI:** 10.3390/jfmk6040091

**Published:** 2021-10-29

**Authors:** Joshua C. Carr, Gena R. Gerstner, Caleb C. Voskuil, Joel E. Harden, Dustin Dunnick, Kristin M. Badillo, Jason I. Pagan, Kylie K. Harmon, Ryan M. Girts, Jonathan P. Beausejour, Matt S. Stock

**Affiliations:** 1Neuromuscular Physiology Laboratory, Texas Christian University, Fort Worth, TX 76129, USA; joshua.carr@tcu.edu (J.C.C.); caleb.voskuil@tcu.edu (C.C.V.); 2Department of Medical Education, TCU & UNTHSC School of Medicine, Fort Worth, TX 76129, USA; 3Neuromechanics Laboratory, Department of Human Movement Sciences, Old Dominion University, Norfolk, VA 23529, USA; ggerstne@odu.edu (G.R.G.); jhard042@odu.edu (J.E.H.); 4Department of Health and Physical Education, Arkansas Tech University, Russellville, AR 72801, USA; ddunnick@atu.edu; 5Neuromuscular Plasticity Laboratory, Institute of Exercise Physiology and Rehabilitation Science, University of Central Florida, Orlando, FL 32816, USA; k.badillo22@knights.ucf.edu (K.M.B.); jason.pagan@ucf.edu (J.I.P.); kylie.harmon@ucf.edu (K.K.H.); ryan.girts@ucf.edu (R.M.G.); jonathan.beausejour@ucf.edu (J.P.B.); 6School of Kinesiology and Physical Therapy, University of Central Florida, Orlando, FL 32816, USA

**Keywords:** ultrasonography, extended field of view, muscle size, muscle quality

## Abstract

The amount of experience with ultrasonography may influence measurement outcomes while images are acquired or analyzed. The purpose of this study was to identify the interrater reliability of ultrasound image acquisition and image analysis between experienced and novice sonographers and image analysts, respectively. Following a brief hands-on training session (2 h), the experienced and novice sonographers and analysts independently performed image acquisition and analyses on the biceps brachii, vastus lateralis, and medial gastrocnemius in a sample of healthy participants (*n* = 17). Test–retest reliability statistics were computed for muscle thickness (transverse and sagittal planes), muscle cross-sectional area, echo intensity and subcutaneous adipose tissue thickness. The results show that image analysis experience generally has a greater impact on measurement outcomes than image acquisition experience. Interrater reliability for measurements of muscle size during image acquisition was generally good–excellent (ICC_2,1_: 0.82–0.98), but poor–moderate for echo intensity (ICC_2,1_: 0.43–0.77). For image analyses, interrater reliability for measurements of muscle size for the vastus lateralis and biceps brachii was poor–moderate (ICC_2,1_: 0.48–0.70), but excellent for echo intensity (ICC_2,1_: 0.90–0.98). Our findings have important implications for laboratories and clinics where members possess varying levels of ultrasound experience.

## 1. Introduction

The use of ultrasonography for skeletal muscle imaging in the field of kinesiology is growing. This is likely a result of its affordability, validity, and reliability compared to advanced imaging techniques such as magnetic resonance imaging and computerized tomography [1,2,3,4]. Establishing the reliability of skeletal muscle ultrasound is critical since it is commonly used to assesses skeletal muscle adaptations with exercise training [1,2,5], muscle disuse [1,6], aging [7,8,9,10] and disease [11,12]. Considerable evidence shows that ultrasound measurements demonstrate acceptable intra and interrater reliability [11,13,14,15,16]. However, the extent to which relative experience with ultrasound image acquisition and analyses influences its outcomes is not well defined.

A common scenario within a research laboratory or clinic is that its members possess varying levels of experience with a technique such as ultrasonography. This presents a challenge, as a critical aspect in longitudinal studies or patient evaluation relates to the feasibility of a single experimenter performing all ultrasound scans and analyses. The fact that relatively minor changes in probe orientation [17], pressure applied to the skin [18], or even scale calibration [16] during analysis can have marked effects on the outcomes illustrates the need to identify interrater reliability between vastly different ultrasound experience levels. Mayer et al. [11] have recently shown that following 8 h of expert-led ultrasound training, a small group of ultrasound-naïve physical therapy students had reliable ultrasound measurements compared to an expert sonographer. Importantly, the scans were performed on a group of patients within an intensive care unit, a group that had recovered from intensive care, and a healthy control, but the group sample sizes were small (*n* = 6), several raters (*n* = 5) of varying experience levels performed acquisition, and the influence of ultrasound analysis experience on the outcomes was not determined. Clearey et al. [19] recently show excellent interrater image analysis reliability for cross-sectional area, muscle thickness, and echo intensity in a group of novices when images were captured by the same, experienced sonographer. When examining interrater reliability for both image acquisition and analysis between novice and expert sonographers, Zaidman et al. [12] show similar outcomes between sonographers but these data were solely on the echo intensity values. Overall, the evidence suggests that ultrasound-derived measurements of skeletal muscle size and quality exhibit acceptable interrater reliability, yet there is insufficient data on how the relative experience of a sonographer influences both image acquisition and the analysis of muscle cross-sectional area, muscle thickness, subcutaneous adipose tissue thickness, and echo intensity.

The present experiment addresses how experience with ultrasonography influences image outcomes by identifying the interrater reliability for ultrasound image acquisition and the analyses between experienced and novice sonographers. An experienced and a novice sonographer performed image acquisition and the subsequent analyses were performed by an experienced and novice image analyst. Outcome measures consisted of muscle thickness, cross-sectional area, subcutaneous adipose tissue thickness, and echo intensity of three commonly studied muscles in the fields of kinesiology—the biceps brachii, vastus lateralis, and medial gastrocnemius muscles.

## 2. Materials and Methods

### 2.1. Study Design

A cross-sectional study design was used to examine the role of B-mode ultrasonography experience on the reliability of ultrasound-derived measurements of muscle thickness, cross-sectional area, echo intensity, and subcutaneous adipose tissue thickness. During a single visit to the University of Central Florida Institute of Exercise Physiology and Rehabilitation Science, participants underwent ultrasound imaging of the biceps brachii, vastus lateralis, and medial gastrocnemius muscles. An experienced and novice sonographer performed image acquisition and an experienced and novice image analyst performed image analysis. The order of testing between sonographers and muscles was randomized with a random number generator. Participants refrained from exercise for ≥24 h before their laboratory visit. All participants signed their Informed Consent, and this study was approved by the Institutional Review Board for Human Subjects at the University of Central Florida (IRB # STUDY00003175).

### 2.2. Participants

A total of 19 participants volunteered for this study and 17 were retained for analyses. The experienced and novice performed the scans together on the first two participants (one female and one male) as part of the hands-on training. Exclusion criteria were limited to neuromuscular or metabolic disease, a history of stroke, cancer, or heart attack, significant musculoskeletal pain, and use of medications that may impact physical performance. Ten females (age = 21 ± 2 yrs; stature = 167.0 ± 9.6 cm; mass = 61.3 ± 8.3 kg; ethnicity: Caucasian = 6, African American = 1, Hispanic = 3) and seven males (age = 24 ± 3 years; stature = 179.5 ± 6.8 cm; mass = 81.9 ± 7.6 kg; ethnicity: Caucasian = 5, Hispanic = 2) were included in analysis.

### 2.3. Sonographers

An experienced (M.S.) and novice (J.C.) sonographer performed all ultrasound scans with the participants on a treatment table. The novice sonographer had never performed ultrasound measurements and was completely naïve to the methods, procedures, and requisite skills necessary to acquire ultrasound images. A brief custom-made video was crafted by an experienced sonographer (G.G.) on the research team regarding the basics of ultrasound image acquisition (Appendix A). On the day of data collection, before acquisition, the experienced sonographer (M.S.) provided the novice with one-on-one instruction regarding the LOGIQ-E software interface, probe orientation, and scanning tips and pitfalls for the imaged muscles. Hands-on training was then accomplished by having the experienced (M.S.) and novice (J.C.) perform the ultrasound scans together on the first two participants. Following this, the experienced (M.S.) and novice (J.C.) performed all scans independently. In total, the novice (J.C.) had less than two hours of instruction before performing the scans without guidance or instruction. At the time of the experiment, the experienced sonographer (M.S.) had approximately seven years of experience with musculoskeletal sonography in adolescents, adults, and the elderly.

### 2.4. B-Mode Ultrasonography Image Acquisition

All images were taken from the right side of the participants while supine for the biceps brachii and vastus lateralis imaging and prone for the medial gastrocnemius assessment. The images were recorded with a portable B-mode imaging device (GE Logiq E BT12, GE Healthcare, Milwaukee, WI, USA) and a multi-frequency linear array probe (12 L-RS, 5–13 MHz, 38.4 mm field of view, GE Healthcare, Milwaukee, WI, USA) was used for the vastus lateralis, whereas a wideband linear array probe (L8-18i-RS, 4.5-18 MHz, 25 mm field of view, GE Healthcare, Milwaukee, WI, USA) was used for the biceps brachii and medial gastrocnemius muscles. All settings were kept consistent (Frequency 10 MHz, Gain 55 dB, Dynamic range 72, Depth 5 cm) across and within participants; however, a depth of 6 cm was required for three participants to view the full muscle and was kept constant across sonographers. Once the site was identified for the respective muscle, sharpie was applied to the skin surface before image acquisition and both sonographers used the identified site for probe placement. For each muscle, still images were captured in the sagittal and transverse planes, and then panoramic images were captured with the panoramic function (LogiqView, GE Healthcare, Milwaukee, WI, USA). Three images were captured for each scan for every muscle. For the panoramic images, the probe was oriented in the transverse plane and was guided by a flexible high-density foam pad to allow steady transverse movement of the probe across the imaging areas. For the biceps brachii, cloth tape was used to identify the 50% distance from the acromion process to the antecubital space. Similarly, the 50% distance between the greater trochanter and the superior border of the patella was used for the vastus lateralis. The site for the medial gastrocnemius muscle was determined on an individual-by-individual basis due to the large heterogeneity of the lower limb [20,21]. The site was identified by scanning in the transverse and sagittal planes of the muscle and visually identifying the site with the largest muscle thickness. A considerable amount of water-soluble transmission gel (Aquasonic 100 ultrasound transmission gel, Parker Laboratories, Inc., Fairfield, NJ, USA) was applied to the skin for all imaging.

### 2.5. Image Analysts

An experienced (G.G.) and novice (C.V.) image analyst were not present during data collection, therefore, were blind to the image coding and were unaware of who acquired the images. A brief custom-made video was crafted by an experienced sonographer (G.G.) on the research team regarding the basics of ultrasound image analyses (Appendix A). The experienced analyst (G.G.) instructed the novice on the procedures for the image analyses, addressing important steps and common challenges for the derived measurements. Hands-on training was then accomplished by having the experienced (G.G.) and novice (C.V.) perform the ultrasound analyses together on the first two participants. Following this, the experienced (G.G.) and novice (C.V.) performed all analyses independently. In total, the novice (C.V.) had less than two hours of instruction before analyzing the images without guidance or instruction. At the time of the experiment, the experienced analyzer (G.G.) had approximately six years of experience with musculoskeletal ultrasound image analyses in adolescents, adults, and the elderly.

### 2.6. B-Mode Ultrasonography Image Analysis

The ultrasound images were exported and analyzed with ImageJ software (version 1.53k; National Institutes of Health, Bethesda, MD, USA). The experienced analyst (G.G.) visually inspected the three images taken for each muscle and site (e.g., biceps brachii sagittal) and selected the clearest image of the three for analysis. The same selected image was then analyzed by the experienced (G.G.) and novice (C.V.) image analysts. The images were first scaled from pixels to cm using the straight-line function. Muscle thickness, in both sagittal and transverse plane, was quantified using the straight-line function at the midpoint of the muscle on the freeze-frame image. To quantify muscle cross-sectional area (cm^2^), the polygon function was used to outline the border of each muscle without any surrounding fascia on the panoramic image. Echo intensity was determined via gray-scale analysis using the histogram function within the same polygon used for cross-sectional area analyses. Using the same image that muscle cross-sectional area and echo intensity were outlined on, subcutaneous adipose tissue thickness was quantified using the straight-line function at three sites (medial, midpoint, lateral) from the skin to the superficial aponeurosis and calculated as the average of the three values.

### 2.7. Statistical Analysis

Descriptive statistics have been reported as the mean ± SD for the following five variables: (1) muscle thickness in the sagittal plane, (2) muscle thickness in the transverse plane, (3) cross-sectional area, (4) echo intensity, and (5) subcutaneous adipose tissue thickness. Paired samples *t*-tests were performed to examine systematic variability, with an alpha level of 0.05 used to determine statistically significant differences. To provide insight into the precision and magnitude of the estimated differences, 95% confidence intervals (CIs) and Cohen’s *d* effect sizes were computed, respectively. Cohen’s *d* values of 0.2, 0.5, and 0.8 were used to classify small, moderate, and large differences, respectively [22]. The method of Bland and Altman [23] was used to identify the 95% limits of agreement between the experienced versus novice sonographers and image analysts. Reliability was quantified with intraclass correlation coefficients (ICCs) and computed with the 2-way random-effects model (ICC_2,1_) on account of its generalizability to other laboratories and testers [24,25]. The ICCs were evaluated based on a reliability scale where ICCs < 0.50 indicated “poor” reliability, ICCs of 0.50–0.75 indicated “moderate” reliability, ICCs of 0.75–0.90 indicated “good” reliability, and ICCs > 0.90 were indicative of “excellent” reliability [26]. The mean square error was used to calculate the absolute standard error of the measurement (SEM [expressed in absolute units and as a percentage of the grand mean]), and the minimal difference needed to be considered real (MD) statistics [25].

## 3. Results

### 3.1. Image Acquisition

Table 1 shows the mean ± SD and % mean difference values for each of the five variables across the three different muscles. Generally, the images acquired by the novice sonographer resulted in larger thickness and cross-sectional area values, with mean differences ranging from 0.38 to 26.47%. Table 2 displays the reliability statistics. Statistically significant differences (*p* < 0.05) were observed for images acquired by the experienced versus novice sonographer for vastus lateralis echo intensity (*p* = 0.002), medial gastrocnemius cross-sectional area (*p* = 0.035), and biceps brachii subcutaneous tissue thickness (*p* = 0.002), and these associated effect sizes were considered moderate or large. All non-significant differences were also associated with small or trivial effect sizes (*d* ≤ 0.456). Based on the ICCs, all variables were classified as demonstrating good–excellent reliability except medial gastrocnemius echo intensity (0.643) and biceps brachii echo intensity (0.437), and subcutaneous tissue thickness (0.740). Variables showing particularly poor SEMs included vastus lateralis subcutaneous tissue thickness (12.38%), biceps brachii sagittal thickness (11.97%), and subcutaneous tissue thickness (17.74%). Figure 1 shows example data, highlighting differences between sonographers.

### 3.2. Image Analysis

Table 3 shows the mean ± SD and % mean difference values for each of the five variables across the three different muscles. Qualitatively, there were no consistent patterns of lower or greater values being demonstrated across variables or muscles. The mean differences ranged from 0.04 to 14.11%. Table 4 displays the reliability statistics. Statistically significant differences (*p* < 0.05) were observed for images analyzed by the experienced versus novice investigator for vastus lateralis cross-sectional area (*p* = 0.005) and biceps brachii transverse thickness (*p* = 0.029). Only vastus lateralis cross-sectional area demonstrated an effect size that was considered moderate (*d* = 0.514). All other effect sizes were considered small (*d* ≤ 0.393). Based on the ICCs, five of the variables demonstrated moderate reliability and one variable showed poor reliability. Eight out of the 15 variables demonstrated SEMs ≥ 10.0%. Figure 2 shows example data, highlighting differences between image analysts.

## 4. Discussion

This experiment describes how the relative experience with ultrasound image acquisition and analyses influences outcomes of skeletal muscle size and quality. The findings from this study show that experience with image acquisition and analysis generally has small effects (*d* < 0.30) and good–excellent (ICC_2,1_ > 0.80–0.98) interrater reliability on most ultrasound outcomes. However, significant differences and large effects between experienced and novice sonographers were observed for image acquisition and analysis for some of the variables. For image acquisition, medial gastrocnemius cross-sectional area, vastus lateralis echo intensity, and biceps brachii subcutaneous adipose tissue thickness were significantly different between sonographers. Despite this, experienced–novice interrater reliability for measures of muscle size was good–excellent (ICC_2,1_ > 0.82–0.98). Similarly, for analysis, vastus lateralis cross-sectional area and biceps brachii muscle thickness were significantly different between sonographers, and measures of muscle size for the vastus lateralis exhibited poor reliability (ICC_2,1_ < 0.65), yet interrater reliability for echo intensity was excellent (ICC_2,1_ > 0.90) for all muscles. The present data show that relative experience with ultrasound techniques has a task-specific influence on image outcomes that should be considered when designing and interpreting ultrasound-based assessments.

There have been limited attempts to quantify the influence of ultrasound experience on image outcomes [11,12,19]. The rationale for this comparison is based on the fact that a laboratory or clinic has a continual rotation of proficiency levels across time. Identifying reliability between high and low experience levels is necessary for study design and coordination. The present findings generally agree with similar reliability studies that have compared individuals with extensive versus limited ultrasound experience [11,12,19]; however, we report greater systematic variability between experienced versus novice sonographers than previously shown [11,12]. The strengths of the present data are that we show how ultrasound acquisition and analysis experience influences the outcomes for the variables used to determine muscle size and quality—muscle cross-sectional area, muscle thickness, subcutaneous adipose thickness, and echo intensity. The poor interrater reliability for echo intensity during image acquisition is likely an artifact of differences in angle, placement, and possibly the speed of the ultrasound probe during acquisition. Whereas the poor interrater reliability for vastus lateralis and biceps brachii muscle size analyses is likely explained by the inability of the novice to discern fascial borders due to their limited experience and challenging shapes of these muscles. As such, the level of skill that is required to accurately acquire ultrasound images may be muscle and variable dependent.

### 4.1. Muscle Cross-Sectional Area

Ultrasound-derived measurements of muscle cross-sectional area have been cross-validated against MRI and CT imaging and show good agreements [2,3,4]. It has been suggested that sonographer proficiency is needed when collecting panoramic images with the extended field of view technique to acquire high-quality images [2]. Indeed the degree of muscle curvature can present challenges and data show that reliability weakens at regions with greater relative curvature, such as the distal portion of the thigh [2,3]. The present data support these observations as the medial gastrocnemius shows greater systematic variability compared to the biceps brachii and vastus lateralis. Nevertheless, excellent interrater reliability is shown for the vastus lateralis (ICCs > 0.90) [2,3,4], the medial gastrocnemius (ICCs > 0.90) [5], and the biceps brachii (ICCs > 0.90) [11] muscles. Our findings extend these observations by showing that a novice sonographer can acquire reliable extended field of view images in these muscles, but experienced–novice disparity increases with technical demand likely due to greater anatomical contour. Interestingly, the experienced–novice comparison for image acquisition versus image analysis shows that reliability was substantially weaker for image analysis. This is an important finding because it demonstrates that extended field of view image analyses requires sonographer proficiency in addition to the skills required for high-quality image acquisition. The present data show that the minimal differences for image acquisition (MD = 2.94, 2.54, 2.76 cm^2^) were smaller than those of analysis (MD = 12.97, 7.85, 3.08 cm^2^) for the vastus lateralis, biceps brachii, and medial gastrocnemius, respectively. There are three critical points to consider based on these data. One, the SEM for experienced–novice image acquisition (0.92–1.06 cm^2^) is similar to that shown for the SEM of ultrasound compared to MRI (0.87 cm^2^) [4] and CT (0.1–1.1 cm^2^) [2]. Two, the minimal differences for experienced–novice image acquisition are likely small enough to detect resistance training-induced muscle hypertrophy and disuse induced atrophy for the lower limb [1,2,4]. Finally, three, the low level of interrater reliability for image analysis would have been unable to detect those effects. Similar outcomes have recently been shown for image analysis interrater reliability in novice sonographers [19]. Despite excellent ICC values, systematic variability was evident for measures of muscle size for the vastus lateralis, rectus femoris, and first dorsal interosseus among three different novice sonographers [19]. Collectively, it seems that experience level affects extended field of view image analysis more than image acquisition.

### 4.2. Echo Intensity and Subcutaneous Adipose Tissue Thickness

The grayscale analysis that determines skeletal muscle echo intensity values is affected by relatively minor alterations in probe positioning mechanics [17]. Given this, it is unsurprising that interrater reliability for echo intensity during acquisition was generally poor–moderate and weaker than the interrater reliability values for image analysis. Zaidman et al. [12] suggest that minimal training is necessary to acquire reliable and clinically valid measures of echo intensity. The authors show that following only a 20-min expert-led training session, interrater reliability for echo intensity between a novice and expert was highly reliable (ICC ≥ 0.85). It is important to point out that echo intensity was derived from polygon tracing of panoramic images in the present study [19], not the rectangle function from still images [11,12]. This distinction likely explains the differing levels of interrater reliability for image acquisition between experience levels in the present study and others [11,12]. The interrater reliability values of the present study for image analysis are similar to those reported for three novice sonographers [19]. The experienced–novice difference in subcutaneous adipose tissue thickness for the biceps brachii during image acquisition is additional evidence that probe mechanics differed between sonographers, emphasizing the importance of probe pressure and alignment during image acquisition. Although the reliability statistics for echo intensity during image acquisition were weaker than those for analysis, the values for the vastus lateralis are comparable to test–retest reliability measures performed by the same, experienced sonographer [6]. Nevertheless, combining the observations of experienced–novice differences in subcutaneous thickness with the comparatively higher SEM and MD values for echo intensity during image acquisition versus analysis, it seems that a novice sonographer requires more than minimal training to acquire, but not analyze, reliable measures of echo intensity.

### 4.3. Muscle Thickness

Jenkins et al. [14] suggest that given the greater skill required for extended field of view imaging, transverse imaging may be a more convenient option for measurements of muscle size. In support, the authors show excellent reliability for cross-sectional area, muscle thickness, and echo intensity determined from panoramic and a single transverse image for the biceps brachii with strong a strong association between muscle cross-sectional area and muscle thickness (r = 0.93). The interrater image acquisition reliability values for muscle thickness obtained in the transverse and sagittal planes show good–excellent reliability with minimal difference values sensitive enough to detect resistance training-induced increases in muscle thickness following longer (>6 weeks) training interventions [5,27,28], but likely not short-term training induced hypertrophy [29]. The SEM for muscle thickness measurements during acquisition are similar to those shown by a single experienced sonographer [30] and those by Mayer et al. [11] with raters of different experience level. Like the issues encountered with muscle cross-sectional area, the interrater reliability for image analysis was considerably weaker for the vastus lateralis and biceps brachii muscles compared to image acquisition. This was not the case for the medial gastrocnemius muscle, likely due to the brightness and clarity of the fascial borders for this muscle compared to the others in this study. Figure 2 shows how the image analysts differed in their muscle size measurements for vastus lateralis cross-sectional area and biceps brachii muscle thickness. The inability of the novice image analyst to consistently identify muscle boundaries and trace faint fascial borders is likely a major factor contributing to the poor interrater reliability for muscle size during image analyses.

### 4.4. Ultrasound Considerations for Novice Onboarding

We recommend that laboratories use structured onboarding procedures for the novice sonographer. We recommend the following considerations for the novice sonographer during onboarding.

Laboratory standards: a video demonstrating image acquisition and analysis procedures for the specific equipment provides an accessible and convenient means to standardize laboratory procedures, techniques, and instructions for the novice (Appendix A). This should be followed up with hands-on experiential learning practices on different skeletal muscles and persons with an experienced sonographer.

Demonstrations: how probe mechanics [16,17,18] influence the ultrasound image should be emphasized during acquisition training. Representative images demonstrating the fascial borders should be provided to the novice during analysis training for the skeletal muscles of interest and their surrounding anatomy, particularly challenging images with faint fascial borders. Since extended field of view scanning and analyses are more technically demanding, these scans should be emphasized and integrated.

Time: the total training time for novice onboarding is challenging to recommend. The formal training time for the present study was ~2 h, whereas ~8 h of training was performed by Mayer et al. [8] who show greater interrater reliability for image acquisition in a more heterogeneous sample. Regardless, recording the amount of time spent training for the novice provides an objective way to monitor their experience. To this point, keeping a formal time log for everyone may be a convenient method to quantify experience levels within a laboratory or clinic.

### 4.5. Limitations

The present data have some important limitations to consider. First, although the sonographers performed the scans independently, the scans were performed on the same site that was determined (and marked) by the first sonographer. It may be that individuals of differing experience would acquire images from different relative muscle locations, yet this was not entirely the question we were attempting to answer. Another limitation is that we did not capture and compare other ultrasound-derived measurements such as fascicle length and pennation angle between novice and experienced sonographers. However, with the rise in automated analysis methods for these measurements, comparisons between an experienced sonographer and the automated analysis are needed [31]. Lastly, although the present sample of participants varied in their anthropometry, training status, and ethnicity, they were all young adults (<27 years) free of disease, illness, and injury. Future studies are encouraged to identify interrater reliability across a longitudinal intervention (i.e., resistance training, disuse, aging) to describe whether different sonographers capture the same magnitude of the respective effect.

## 5. Conclusions

The present data show how experience with ultrasound image acquisition and analysis influences measurements of skeletal muscle size and quality. Since ultrasound imaging is a relatively simple procedure, it offers a lower barrier to entry for skeletal muscle measurements compared to other techniques. This has spurred interest in the question, how much training is required for ultrasound proficiency? Despite the level of convenience for ultrasound image acquisition and analysis, our data show that a tradeoff exists for interrater reliability between experienced and novice sonographers during image acquisition versus analyses. The experienced–novice comparisons for image acquisition show measures of muscle size can be reliably acquired, but measurements of muscle quality cannot. Whereas experienced–novice comparisons for image analyses indicate that measures of muscle quality are reliably analyzed, but measurements of muscle size for the vastus lateralis and biceps brachii are not. Many authors have shown that skill is required for high-quality ultrasound image acquisition, yet a critical interpretation from the present study is that ultrasound image analyses are not trivial procedures. Comparatively speaking, the experienced–novice differences were more severe for image analyses than image acquisition. Collectively, these findings suggest that ultrasound image analysis experience has a greater influence on the derived outcome variables than acquisition experience. The findings of this study have implications for laboratories that use ultrasonography and possess members of varying experience levels.

## Figures and Tables

**Figure 1 jfmk-06-00091-f001:**
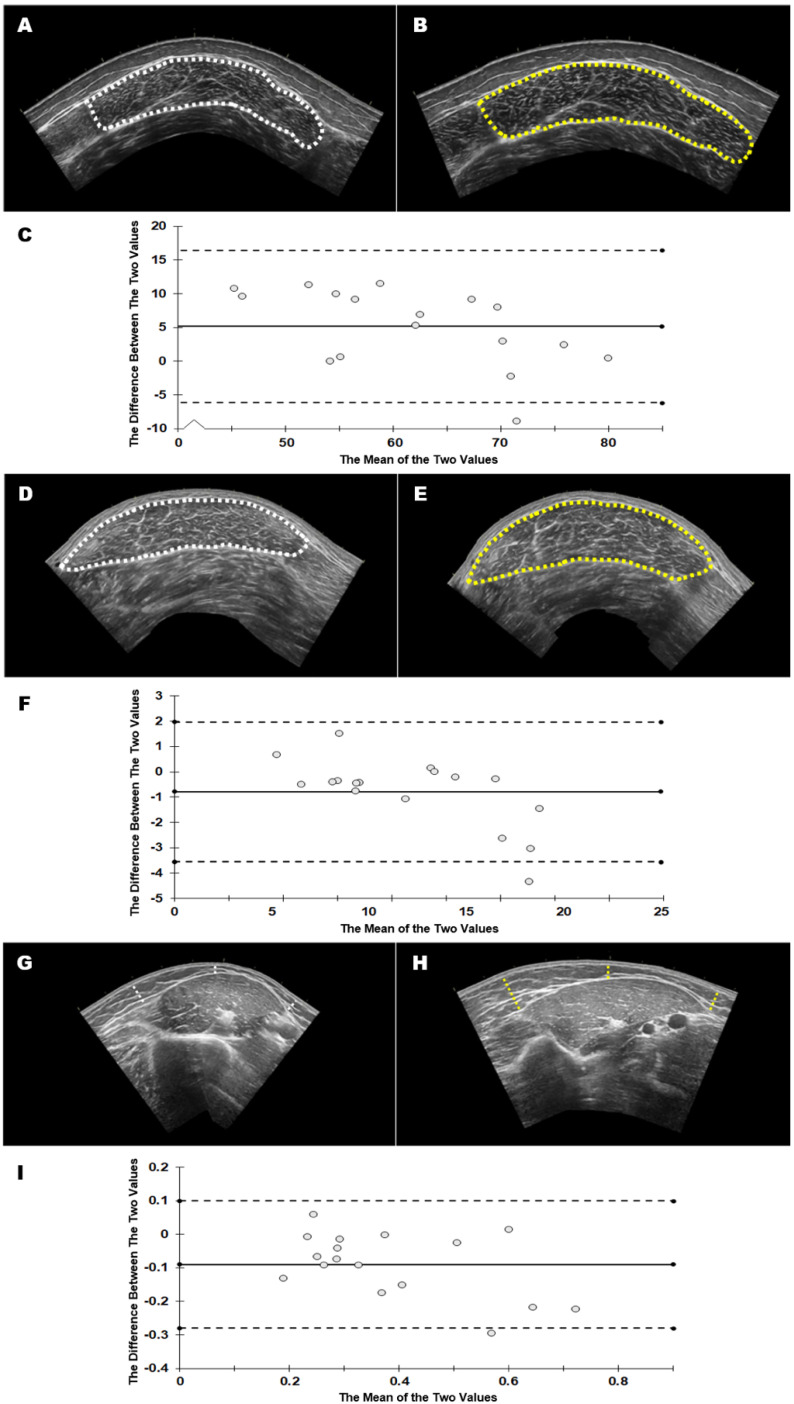
Acquired images from the experienced sonographer (left column, **A**,**D**,**G**) and the novice sonographer (right column, **B**,**E**,**H**) were analyzed by the experienced analyst (G.G.). The variables that were significantly different between sonographers during acquisition are shown: echo intensity of the vastus lateralis (top row, **A**,**B**), cross sectional area of the medial gastrocnemius (middle row, **D**,**E**), and subcutaneous adipose tissue thickness of the biceps brachii (bottom row, **G**,**H**). Bland Altman plots are shown for each variable (**C**,**F**,**I**). Dotted lines: 95% limits of agreement.

**Figure 2 jfmk-06-00091-f002:**
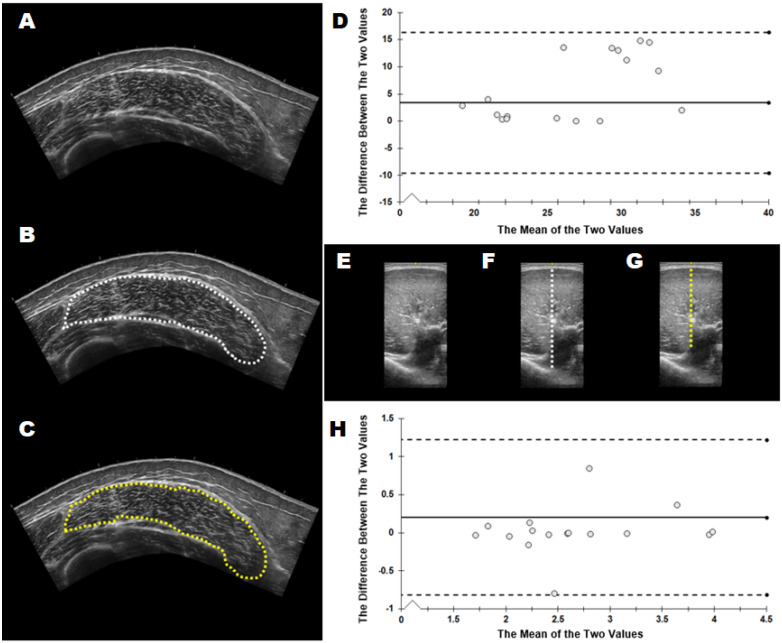
Acquired images (**A**,**E**) from the experienced sonographer (M.S.) were analyzed by the experienced analyst (**B**,**F**) and the novice analyst (**C**,**G**). Representative images for the variables that were significantly different between analysts during image analysis are shown: cross sectional area of the vastus lateralis (**B**,**C**) and transverse thickness of the biceps brachii (**F**,**G**). Bland Altman plots are shown for vastus lateralis cross-sectional area (**D**) and biceps brachii thickness (**H**). Dotted lines: 95% limits of agreement.

**Table 1 jfmk-06-00091-t001:** Mean ± SD values and the % mean difference between the experienced and novice sonographers for each muscle and variable.

	Transverse Thickness (cm)	Sagittal Thickness (cm)	CSA (cm^2^)	EI (A.U.)	Subcutaneous Thickness (cm)
**Vastus Lateralis**					
Experienced	2.39 ± 0.52	2.38 ± 0.54	27.04 ± 9.23	64.48 ± 8.96	1.17 ± 0.65
Novice	2.49 ± 0.41	2.46 ± 0.45	26.74 ± 8.65	59.32 ± 12.09	1.22 ± 0.62
% Mean Difference	4.18	3.36	1.11	8.00	4.27
**Medial Gastrocnemius**					
Experienced	1.86 ± 0.41	1.84 ± 0.36	11.74 ± 3.98	110.56 ± 9.84	0.585 ± 0.240
Novice	1.90 ± 0.38	1.90 ± 0.42	12.53 ± 4.93	110.14 ± 9.49	0.587 ± 0.248
% Mean Difference	2.15	3.26	6.73	0.38	1.72
**Biceps Brachii**					
Experienced	2.61 ± 0.77	2.44 ± 0.77	9.65 ± 5.54	120.87 ± 8.93	0.341 ± 0.144
Novice	2.79 ± 0.78	2.64 ± 0.92	10.00 ± 6.19	123.53 ± 11.87	0.431 ± 0.192
% Mean Difference	6.90	8.20	3.63	2.20	26.47

**Table 2 jfmk-06-00091-t002:** Statistical outcomes for the image acquisition comparisons between the experienced versus novice sonographers. * = statistically significant difference between sonographer images.

	Transverse Thickness (cm)	Sagittal Thickness (cm)	CSA (cm^2^)	EI (A.U.)	Subcutaneous Thickness (cm)
**Vastus Lateralis**					
*p*	0.136	0.288	0.417	0.002 *	0.334
*d*	0.381	0.267	0.202	0.898	0.241
95% CI	−0.237–0.035	−0.203–0.066	−0.468–1.074	2.208–8.123	−0.158–0.057
Limits of Agreement	−0.621–0.419	−0.594–0.452	−2.637–3.243	−6.110–16.440	−0.461–0.360
Upper–Lower Limit					
ICC_2,1_	0.826	0.854	0.986	0.77	0.945
SEM (raw units)	0.19	0.19	1.06	4.07	0.15
SEM (%)	7.69	7.79	3.95	6.57	12.38
MD	0.52	0.52	2.94	11.27	0.41
**Medial Gastrocnemius**					
*p*	0.226	0.105	0.035 *	0.838	0.92
*d*	0.305	0.416	0.559	0.05	0.025
95% CI	−0.118–0.030	−0.137–0.014	−1.512–−0.063	−3.859–4.697	−0.039–0.035
Limits of Agreement	−0.325–0.238	−0.351–0.228	−3.551–1.976	−15.888–16.727	−0.143–0.149
Upper–Lower Limit					
ICC_2,1_	0.932	0.92	0.939	0.643	0.959
SEM (raw units)	0.10	0.10	1.00	5.88	0.05
SEM (%)	5.41	5.58	8.22	5.33	8.71
MD	0.28	0.29	2.76	16.31	0.14
**Biceps Brachii**					
*p*	0.082	0.078	0.28	0.341	0.002 *
*d*	0.45	0.456	0.271	0.238	0.926
95% CI	−0.365–0.024	−0.418–0.025	−1.019–0.315	−8.394–3.078	−0.139–0.040
Limits of Agreement	−0.913–0.571	−1.040–0.647	−2.894–2.190	−24.524–19.207	−0.279–0.100
Upper–Lower Limit					
ICC_2,1_	0.867	0.856	0.975	0.437	0.74
SEM (raw units)	0.27	0.30	0.92	7.89	0.07
SEM (%)	9.91	11.97	9.34	6.46	17.74
MD	0.74	0.84	2.54	21.87	0.19

**Table 3 jfmk-06-00091-t003:** Mean ± SD values and the % mean difference between the experienced and novice image analysts for each muscle and variable.

	Transverse Thickness (cm)	Sagittal Thickness (cm)	CSA (cm^2^)	EI (A.U.)	Subcutaneous Thickness (cm)
**Vastus Lateralis**					
**Experienced**	2.44 ± 0.46	2.42 ± 0.49	26.89 ± 8.81	61.90 ± 10.80	1.20 ± 0.62
**Novice**	2.58 ± 0.54	2.58 ± 0.54	23.49 ± 8.12	61.98 ± 11.52	1.12 ± 0.52
% **Mean Difference**	5.95	6.78	12.64	0.13	6.40
**Medial Gastrocnemius**					
**Experienced**	1.87 ± 0.39	1.87 ± 0.38	12.13 ± 4.43	110.35 ± 9.52	0.586 ± 0.241
**Novice**	1.91 ± 0.43	1.92 ± 0.45	11.67 ± 4.36	110.84 ± 9.46	0.568 ± 0.254
% **Mean Difference**	1.47	2.47	3.84	0.44	3.10
**Biceps Brachii**					
**Experienced**	2.70 ± 0.77	2.54 ± 0.84	9.82 ± 5.78	122.20 ± 10.43	0.386 ± 0.173
**Novice**	2.50 ± 0.64	2.55 ± 0.75	8.44 ± 3.94	122.25 ± 10.84	0.399 ± 0.210
% **Mean Difference**	7.55	0.20	14.11	0.04	3.45

**Table 4 jfmk-06-00091-t004:** Statistical outcomes for the image analysis comparisons between the experienced versus novice analysts. * = statistically significant difference between image analysts.

	Transverse Thickness (cm)	Sagittal Thickness (cm)	CSA (cm^2^)	EI (A.U.)	Subcutaneous Thickness (cm)
**Vastus Lateralis**					
*p*	0.087	0.07	0.005 *	0.827	0.222
*d*	0.303	0.321	0.514	0.038	0.214
95% CI	−0.312 to 0.022	−0.342 to 0.014	1.090 to 5.709	−0.807 to 0.650	−0.049 to 0.202
Limits of Agreement	−1.084 to 0.794	−1.165 to 0.837	−9.572 to 16.371	−4.171 to 4.014	−0.626 to 0.780
Upper–Lower Limit					
ICC_2,1_	0.527	0.489	0.648	0.983	0.801
SEM (raw units)	0.34	0.36	4.68	1.48	0.25
SEM (%)	13.5	14.4	18.6	2.4	21.9
MD	0.94	1.00	12.97	4.09	0.7
**Medial Gastrocnemius**					
*p*	0.313	0.178	0.093	0.307	0.158
*d*	0.176	0.236	0.296	0.178	0.248
95% CI	−0.083 to 0.027	−0.115 to 0.022	−0.082 to 1.015	−1.452 to 0.471	−0.007 to 0.044
Limits of Agreement	−0.337 to 0.281	−0.430 to 0.338	−2.615 to 3.547	−5.890 to 4.910	−0.125 to 0.162
Upper–Lower Limit					
ICC_2,1_	0.927	0.887	0.933	0.958	0.955
SEM (raw units)	0.11	0.14	1.11	1.95	0.05
SEM (%)	5.9	7.3	9.3	1.8	9.0
MD	0.31	0.38	3.08	5.4	0.14
**Biceps Brachii**					
*p*	0.029 *	0.952	0.052	0.953	0.486
*d*	0.393	0.01	0.346	0.01	0.121
95% CI	0.023 to 0.385	−0.175 to 0.165	−0.012 to 2.783	−1.675 to 1.579	−0.052 to 0.025
Limits of Agreement	−0.815 to 1.223	−0.960 to 0.950	−6.465 to 9.236	−9.190 to 9.094	−0.229 to 0.202
Upper–Lower Limit					
ICC_2,1_	0.709	0.818	0.653	0.906	0.839
SEM (raw units)	0.37	0.35	2.83	3.3	0.08
SEM (%)	14.1	13.5	31	2.7	19.8
MD	1.02	0.96	7.85	9.14	0.22

## Data Availability

The data presented in this study are available on request from the corresponding author. The data are not publicly available.

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
