# Peer review of "The Influence of Sonographer Experience on Skeletal Muscle Image Acquisition and Analysis"

_jfmk, 2021, doi:10.3390/jfmk6040091_

Round 1

Reviewer 1 Report

The paper by Carr et al. entitled “The influence of sonographer experience on skeletal muscle image acquisition and analysis” aims at identifying the interrater reliability for ultrasound image acquisition and analysis between experienced and novice sonographers. Specifically, experienced and novice sonographers and analysts performed image acquisition and analysis on biceps brachii, vastus lateralis and medial gastrocnemius of healthy patients, both males and females of different ethnicities.

The measurements considered by authors comprised muscle thickness, cross-sectional area, subcutaneous adipose tissue thickness and echo intensity. The authors found that significant differences between experienced and novice sonographers were found in both image acquisition and analysis, as indicated by asterisks (= statistically significant difference).

Authors affirmed that the level of training and experience is definitely important for the acquisition and analysis of ultrasound images. Specifically, they found that the experience level is more important in image analysis than in image acquisition, although they also affirmed that novice sonographers require extended training to acquire, but not analyze, accurate measurements of echo intensity, which for instance could be affected by probe pressure and alignment, as also suggested by Figure 1. Given this considerations, this paper shed light to the importance of the experience needed in ultrasound image acquisition and analysis. This founding has been achieved with a methodic and accurate experimental procedure.

Overall, the paper is well written and in my opinion pretty comprehensible also to researchers/scientists that are not into this field, by the way I think that few points have to be addressed:

  1. Line 72: Instead of using “this” I suggest the authors to briefly repeat what they mean, so that the reader does not lose the focus of the subject
  2. Line 138: It is not totally clear if “each” is referred to “each muscle” or “each participant”. I would suggest to specify

Table captions: the table captions are not completely clear. I would suggest to implement the description with more details.

Reviewer 2 Report

This manuscript investigated agreement in ultrasound based measurements of skeletal muscle between novice and experienced monographers and data analysts. Overall the paper is well written, motivated, and executed. The statistics were performed correctly. There is one small issue that should be addressed, otherwise the manuscript is ready for publication.

The authors claim, "When designing the study, we sought to enroll a heterogeneous sample of participants to provide the sonographers and image analysts with varying degrees of difficulty" However it seems like the population that was enrolled was fairly young and healthy. With age there is generally a decrease in muscle size and an increase in fatty infiltration that would likely make data acquisition and analysis more difficult. Therefore the authors should remove the claim that this is a heterogenous population and make sure to claim that this is a young, healthy population in the limitations section.
